# Effectiveness and safety of rivaroxaban versus warfarin in Taiwanese patients with end-stage renal disease and nonvalvular atrial fibrillation: A real-world nationwide cohort study

Yi-Cheng Lin[1,2], Bi-Li Chen[1,2], Chun-Ming Shih[3,4,5,6], Feng-Yen Lin[3,4,5,6], Chih-Wei Chen[3,4,5,6], Chien-Yi Hsu[3,4,5,6], Yung-Ta Kao[3,4,5,6,7], Wei-Fung Bi[3,4,5,6], Li-Ying Chen[8], Li-Nien Chien[9]☯*, Te-Chao Fang[10,11,12]☯*, Chun-Yao Huang[3,4,5,6]☯*

1 Department of Pharmacy, Taipei Medical University Hospital, Taipei, Taiwan, 2 School of Pharmacy, College of Pharmacy, Taipei Medical University, Taipei, Taiwan, 3 Division of Cardiology, Department of Internal Medicine, Taipei Medical University Hospital, Taipei, Taiwan, 4 Cardiovascular Research Center, Taipei Medical University Hospital, Taipei, Taiwan, 5 Taipei Heart Institute, Taipei Medical University, Taipei, Taiwan, 6 Division of Cardiology, Department of Internal Medicine, School of Medicine, College of Medicine, Taipei Medical University, Taipei, Taiwan, 7 Professional Master Program in Artificial Intelligence in Medicine, College of Medicine, Taipei Medical University, Taipei, Taiwan, 8 Health and Clinical Research Data Center, Office of Data Science, Taipei Medical University, Taipei, Taiwan, 9 School of Health Care Administration, College of Management, Taipei Medical University, Taipei, Taiwan, 10 Division of Nephrology, Department of Internal Medicine, Taipei Medical University Hospital, Taipei Medical University, Taipei, Taiwan, 11 Division of Nephrology, Department of Internal Medicine, School of Medicine, College of Medicine, Taipei Medical University, Taipei, Taiwan, 12 TMU Research Center of Urology and Kidney, Taipei Medical University, Taipei, Taiwan

☯ These authors contributed equally to this work.
* cyhuang@tmu.edu.tw (CYH); lnchien@tmu.edu.tw (LNC); fangtechao@gmail.com (TCF)

**Data Availability Statement:** In regards to data availability, our study used National Health

## Abstract

### Background

The optimal anticoagulant for end-stage renal disease patients for stroke prophylaxis is unknown. The efficacy and safety of warfarin in this population are debatable. In addition, real-world evidence of direct oral anticoagulants in patients with end-stage renal disease is limited. The aim of this study was to evaluate the clinical outcomes of rivaroxaban compared with warfarin in Taiwanese patients with end-stage renal disease with nonvalvular atrial fibrillation in a real-world setting.

### Methods and results

This was a retrospective population-based cohort study conducted using Taiwan's National Health Insurance Research Database. Patients with nonvalvular atrial fibrillation and end-stage renal disease who started on rivaroxaban or warfarin between February 2013 and September 2017 were eligible to participate in the study. The inverse probability of treatment weighting approach was used to balance baseline characteristics. Bleeding and thrombo-embolic outcomes were compared using competing risk analyses. The study population

Insurance Research Data, a healthcare claims data that provided by the Health and Welfare Science Data Center (HWDC), Ministry of Health and Welfare in Taiwan. The HWDC is a third-party organization. Researchers can submit application to HWDC in order to have access to several health-related databases. Due to legal restrictions imposed by the government of Taiwan in relation to the Personal Information Protection Act, data cannot be made publicly available. Requests for data can be sent as a formal proposal to the HWDC with an IRB approval letter. The contact information of Taipei Medical University Joint IRB is tmujirb@gmail.com. All data were fully anonymized before we access them. In addition, these data can only be access and analyzed in an independent operating area in the HWDC. Only statistical results can be brought out from the operating area. Therefore, original data cannot be shared publicly due to legal restrictions.

**Funding:** The authors received no specific funding for this work.

**Competing interests:** The authors have declared that no competing interests exist.

consisted of 3358 patients (173 and 3185 patients on rivaroxaban and warfarin, respectively). In the rivaroxaban group, 50.8%, 38.7%, and 10.4% of the patients received 10, 15, and 20 mg of the drug, respectively. The cumulative incidence of major bleeding was similar between the two groups; however, the gastrointestinal bleeding rate was lower in the rivaroxaban group (adjusted subdistribution hazard ratio [SHR]: 0.56, 95% confidence interval [CI]: 0.34–0.91) than in the warfarin group. Furthermore, the composite risk of ischemic stroke or systemic embolism was significantly lower in the rivaroxaban group (adjusted SHR: 0.36, 95% CI: 0.17–0.79). Similar findings were observed for patients who received 10 mg of rivaroxaban.

## Conclusions

In Taiwanese patients with end-stage renal disease and nonvalvular atrial fibrillation, rivaroxaban may be associated with a similar risk of major bleeding but a lower risk of thromboembolism compared with warfarin. The potential benefit of 10 mg of rivaroxaban in this population requires further investigation.

## Introduction

Nonvalvular atrial fibrillation (NVAF) is common in patients with chronic kidney disease, and the prevalence markedly increases as renal function declines [1, 2]. An estimated 13%–27% of patients with end-stage renal disease (ESRD) have NVAF [3, 4], a substantially higher prevalence than in the general population. In addition, chronic kidney disease increases the stroke risk independent of other risk factors in patients with NVAF [5]. Despite an increased thromboembolism risk in patients with ESRD and NVAF, anticoagulant use in this population has been controversial because it lacks sufficient benefits, and anticoagulant users have had more adverse effects than nonusers [6, 7]. Moreover, stroke prevention is complex because renal dysfunction is an independent risk factor for major bleeding [1, 8].

To date, the optimal anticoagulant for the ESRD population for stroke prophylaxis is unknown. The efficacy and safety of warfarin in patients with ESRD for stroke prophylaxis are debatable. Numerous observational studies and meta-analyses have suggested that warfarin has no clear benefit and indicated that it is associated with increased bleeding compared with no anticoagulant and direct oral anticoagulant use in patients with ESRD [6, 9–13]. Direct oral anticoagulants have been demonstrated to be beneficial over warfarin in patients with NVAF in phase 3 clinical trials [14–18]. However, patients with ESRD were excluded from these trials, considering that direct oral anticoagulants are primarily eliminated through the kidney and that this population has high mortality and morbidity risks. A recent randomized controlled trial compared the efficacy and safety of apixaban with warfarin for stroke prevention in patients with NVAF and ESRD [19]. However, the trial was stopped early, leaving the results inconclusive. Direct oral anticoagulant use in this population has been investigated using real-world data in the United States, but Caucasians were the large majority in these study populations, and conflicting results were obtained [20–23].

In Taiwan, rivaroxaban is approved for stroke prophylaxis in NVAF patients with creatinine clearance of ≥15 mL/min. In addition to 15 mg of rivaroxaban, 10 mg of rivaroxaban is approved in Taiwan and Japan for patients with creatinine clearance between 15 and 50 mL/min. The approval was based on the findings of a phase 3 randomized controlled trial in Japan

[24], and in that trial, a lower dosage was chosen for investigation based on previous pharmacokinetic data in Japanese patients. The elimination of rivaroxaban is less dependent on renal clearance compared with dabigatran and edoxaban [25–28], which makes it a potential option for patients with severe renal dysfunction. To our knowledge, no real-world data are available regarding the evaluation of the off-label use of rivaroxaban for stroke prophylaxis in Asian patients with ESRD. The study objective was to investigate the effectiveness and safety of rivaroxaban compared with warfarin in Asian patients with NVAF and ESRD in a real-world setting.

## Methods

### Study design and data sources

This was a retrospective population-based cohort study conducted using Taiwan's National Health Insurance Research Database. This database contains insurance claims from 99% of Taiwan residents. The database captures enrollment records; International Classification of Diseases, Ninth and Tenth Revision (ICD-9 and ICD-10) diagnosis codes; procedure codes; and prescription records from both inpatient and outpatient services. This study was approved by the Joint Institutional Review Board of Taipei Medical University (TMU-JIRB No. N201911006). Because all data were de-identified, the Institutional Review Board waived the need for informed consent.

### Study cohort

We used prescriptions records to select a study cohort to minimize the possibility of underreporting and incomplete diagnosis coding because the National Health Insurance Research Database only captured up to five diagnoses for each visit. We selected patients who received oral anticoagulant prescriptions between February 2013 and September 2017. We excluded patients from the cohort if (1) they were aged <20 years; (2) their anticoagulant prescription was filled only once during the study period; (3) anticoagulants were not prescribed by neurologists or cardiologists; (4) they received a diagnosis of pulmonary embolism or deep vein thrombosis within 6 months before the index date; and (5) they received joint replacement or valvular surgery within 6 months before the index date [29]. Among patients on oral anticoagulants with a diagnosis of NVAF or atrial flutter, we selected patients on rivaroxaban or warfarin with an ESRD diagnosis as our final study cohort. ESRD, which was defined based on a diagnosis of stage 5 chronic kidney disease or patients being on regular dialysis in this study, was identified through ICD-9 codes in the Registry of Catastrophic Illness and medical records indicating the use of erythropoiesis-stimulating agents. According to National Health Insurance policies, patients who have received renal replacement therapy for at least 3 months are eligible for catastrophic illness certification, and the use of erythropoiesis-stimulating agents is limited to patients with ESRD regardless of the dialysis status. For erythropoiesis-stimulating drug users, we specifically included those with an ESRD diagnosis identified through ICD-9 codes to eliminate patients using these agents for off-label indications. The patient selection process is shown in Fig 1. The study cohort was followed from the date of the first anticoagulant prescription to the date of the clinical event of interest or until December 31, 2017, whichever came first.

### Comorbidities and medications

Thromboembolic and bleeding risks at the baseline were assessed using established scoring systems, namely the $CHA_2DS_2$-VASc and ORBIT scores. The $CHA_2DS_2$-VASc score

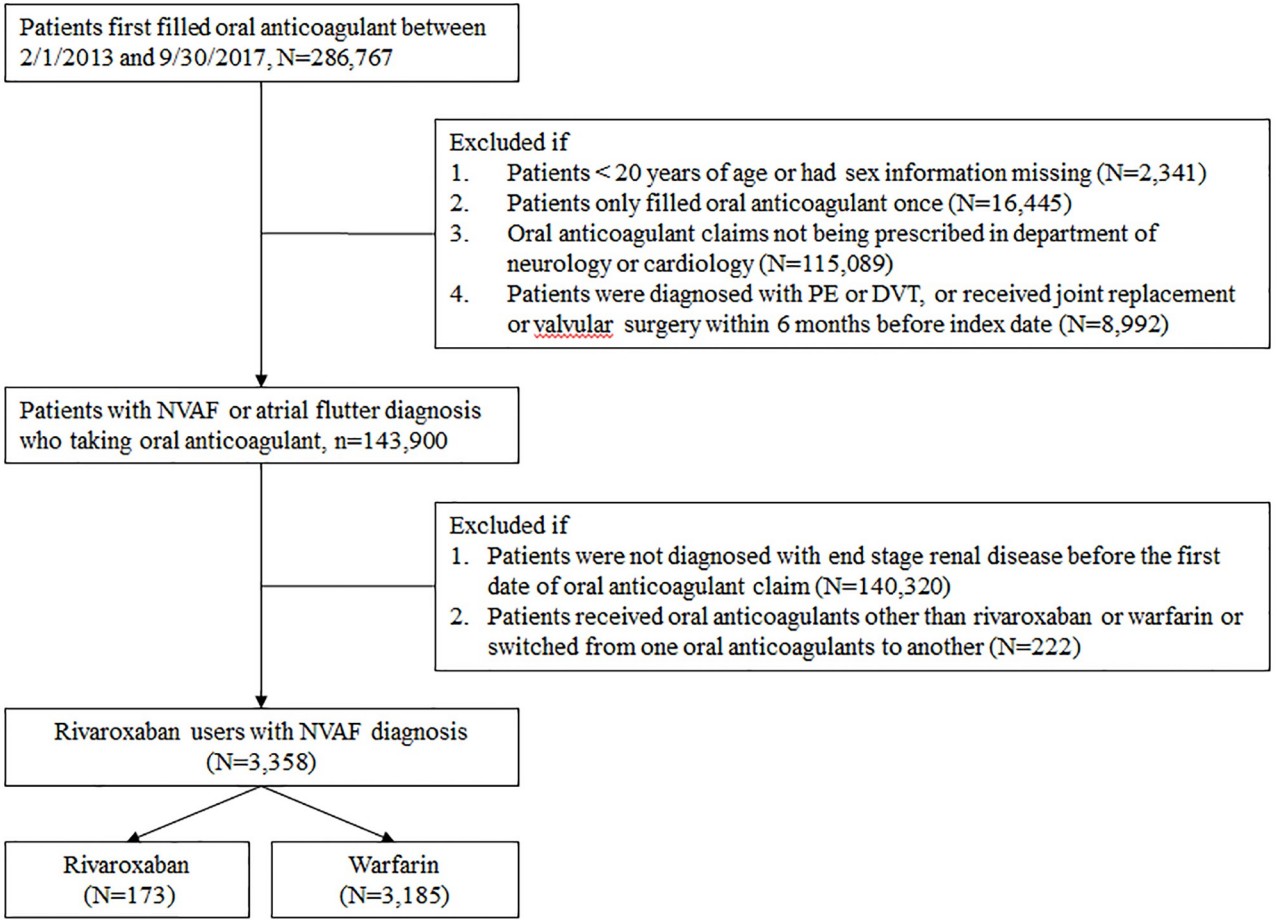

**Fig 1. Patient selection process.** A total of 3358 patients with NVAF and ESRD receiving either rivaroxaban or warfarin were enrolled in this study, consisting of 173 and 3185 rivaroxaban and warfarin users, respectively. NVAF = nonvalvular atrial fibrillation; DVT = deep vein thrombosis; PE = pulmonary embolism.

outperformed the $CHADS_2$ score in predicting thromboembolic risk in the Taiwanese population with NVAF [30]. The ORBIT score had better accuracy than other bleeding risk scoring systems in predicting major bleeding in patients with NVAF and was validated in a large cohort of patients receiving rivaroxaban or warfarin [31]. Specific diagnosis and medication codes for comorbidities and medications are listed in S1 Table.

## Study outcomes

The outcomes of interest are safety and efficacy [29]. Safety outcomes include hospitalization for major bleeding, defined as fatal bleeding, symptomatic bleeding in a critical area or organ, or bleeding leading to transfusions, and non-major clinically relevant bleeding. The definition of major bleeding was based on the recommendations of the International Society on Thrombosis and Haemostasis [32]. Non-major clinically relevant bleeding was defined as any hemorrhage that did not satisfy the criteria for major bleeding but led to hospitalization or medical visits. Efficacy outcomes included the composite endpoint of ischemic stroke or systemic embolism and individual components of the composite endpoint. Study outcomes were identified based on disease diagnosis codes and procedure codes, which are provided in S1 Table. Because 10 mg of rivaroxaban accounted for a large proportion of usage and is only approved

in Taiwan and Japan, we performed analyses comparing clinical outcomes between users of 10 mg of rivaroxaban and warfarin.

## Statistical analysis

To reduce potential selection bias, we used inverse probability of treatment weighting (IPTW) based on the propensity score to balance the baseline characteristics of patients receiving warfarin and rivaroxaban, resulting in similar baseline characteristics between the two groups. Instead of matching two treatment groups based on the selected confounders, IPTW involves using the entire cohort and can address numerous confounding variables. IPTW allows for the estimation of marginal hazard ratios with minimal bias while retaining data from all participants [33, 34]. Each patient was assigned a weight based on the likelihood of exposure to the treatment effect, which was estimated through logistic regression. We considered all baseline characteristics when estimating the weight. Standardized mean difference was used to compare baseline characteristics between the two groups, and a value of <0.1 indicated a negligible difference between the variables of the treatment groups. Because the outcomes of interest were thromboembolic and bleeding events, we considered death as a competing risk. Therefore, the cumulative incidence of competing risk was used to estimate the incidence of the selected outcomes. The adjusted subdistribution hazard ratio (SHR) was calculated using the competing risk model adjusted for sex, age, comorbidities, and prescribed medications. The warfarin group served as the reference cohort. During the follow-up period, patients who switched from rivaroxaban to warfarin and vice versa were excluded. Analyses were performed using SAS/STAT 9.4 software (SAS Institute Inc., Cary, NC, USA) and STATA 14 software (Stata Corp LP, College Station, TX, USA). A p-value of <0.05 was considered statistically significant.

## Results

A total of 286,767 patients were selected from February 2013 to September 2017. Of these, 3358 patients met the eligibility criteria and were included in the analysis (173 and 3185 patients receiving rivaroxaban or warfarin, respectively). In the rivaroxaban group, 88 (50.8%), 67 (38.7%), and 18 (10.4%) patients received 10, 15, and 20 mg of the drug, respectively. The mean follow-up durations for the rivaroxaban and warfarin groups were 19.1 and 27.4 months, respectively. Before IPTW, patients in the rivaroxaban group were older and had more comorbidities than those in the warfarin group. The mean $CHA_2DS_2$-VASc and ORBIT scores of those in the rivaroxaban group were higher than those in the warfarin group ($CHA_2DS_2$-VASc and ORBIT scores were 4.0 vs. 3.7 and 2.9 vs. 2.7, respectively). After IPTW, the baseline characteristics were balanced between the two groups. Detailed baseline characteristics are listed in Table 1.

The cumulative incidence and competing risk of safety outcomes are shown in Fig 2 and Table 2. The major bleeding risk was similar between rivaroxaban and warfarin users (adjusted SHR: 0.86, 95% confidence interval [CI]: 0.50–1.47, p = 0.59). No significant difference was observed in the risk of non-major clinically relevant bleeding (adjusted SHR: 0.74, 95% CI: 0.48–1.13, p = 0.16). We further classified these bleeding events based on bleeding origin. The gastrointestinal bleeding risk was significantly lower in the rivaroxaban group than in the warfarin group (adjusted SHR: 0.56, 95% CI: 0.34–0.91, p = 0.02), whereas the intracranial bleeding risk was similar between the groups (adjusted SHR: 0.62, 95% CI: 0.24–1.61, p = 0.33).

The cumulative incidence and competing risk of efficacy outcomes are shown in Fig 2 and Table 2. The composite risk of ischemic stroke or systemic embolism was significantly lower in the rivaroxaban group than in the warfarin group (adjusted SHR: 0.36, 95% CI: 0.17–0.79,

**Table 1. Baseline characteristics of eligible patients received rivaroxaban and warfarin.**

| | Before IPTW | | | After IPTW | | |
|---|---|---|---|---|---|---|
| | Rivaroxaban (n = 173) | Warfarin (n = 3185) | | Rivaroxaban (n = 173) | Warfarin (n = 3185) | |
| | % | % | SMD | % | % | SMD |
| **Male** | 55 | 51 | 0.08 | 43 | 49 | 0.13 |
| **Age, mean ± SD (y)** | 75 ± 9 | 69 ± 12 | 0.54 | 69 ± 11 | 69 ± 12 | 0.02 |
| 20–64 | 15 | 35 | 0.49 | 39 | 34 | 0.09 |
| 65–74 | 32 | 31 | 0.04 | 29 | 31 | 0.03 |
| 75+ | 53 | 34 | 0.39 | 32 | 35 | 0.06 |
| **Charlson–Deyo index, mean ± SD** | 6 ± 3 | 5 ± 2 | 0.35 | 5 ± 2 | 5 ± 2 | 0.13 |
| 0–2 | 9 | 13 | 0.13 | 7 | 13 | 0.18 |
| 3 | 13 | 18 | 0.16 | 17 | 18 | 0.02 |
| 4+ | 78 | 69 | 0.22 | 76 | 69 | 0.14 |
| **CHA$_2$DS$_2$-VASc score, mean ± SD** | 4.0 ± 1.5 | 3.7 ± 1.6 | 0.18 | 3.8 ± 1.5 | 3.7 ± 1.6 | 0.05 |
| 0–2 | 20 | 25 | 0.12 | 20 | 25 | 0.11 |
| 3 | 17 | 25 | 0.13 | 24 | 22 | 0.03 |
| 4+ | 63 | 52 | 0.21 | 56 | 53 | 0.06 |
| **ORBIT score** | 2.9 ± 1.4 | 2.7 ± 1.4 | 0.13 | 2.8 ± 1.5 | 2.7 ± 1.4 | 0.05 |
| 0–2 | 49 | 55 | 0.14 | 54 | 55 | 0.02 |
| 3 | 22 | 20 | 0.07 | 20 | 20 | 0.004 |
| 4+ | 29 | 25 | 0.09 | 26 | 25 | 0.02 |
| **Comorbidities** | | | | | | |
| Ischemic stroke | 19 | 13 | 0.15 | 16 | 13 | 0.08 |
| GI bleeding | 13 | 12 | 0.03 | 15 | 12 | 0.09 |
| Myocardial infarction | 10 | 7 | 0.09 | 11 | 8 | 0.12 |
| Congestive heart failure | 33 | 37 | 0.08 | 43 | 37 | 0.13 |
| Peptic ulcer disease | 29 | 22 | 0.16 | 24 | 22 | 0.04 |
| Hypertension | 82 | 78 | 0.09 | 78 | 78 | 0.02 |
| Diabetes | 41 | 51 | 0.19 | 51 | 50 | 0.02 |
| Chronic liver disease | 9 | 7 | 0.08 | 6 | 7 | 0.07 |
| Hyperlipidemia | 28 | 23 | 0.11 | 18 | 23 | 0.13 |
| COPD | 16 | 13 | 0.08 | 10 | 13 | 0.10 |
| Valvular heart disease | 11 | 11 | 0.002 | 12 | 11 | 0.04 |
| Malignancy | 47 | 14 | 0.77 | 13 | 16 | 0.08 |
| **Medication history** | | | | | | |
| NSAID | 35 | 27 | 0.17 | 24 | 28 | 0.08 |
| Glucocorticoids | 23 | 15 | 0.20 | 14 | 16 | 0.05 |
| Antiplatelet agents | 57 | 52 | 0.09 | 50 | 53 | 0.06 |
| PPI | 19 | 14 | 0.14 | 16 | 14 | 0.06 |
| HMG-CoA reductase inhibitors | 25 | 21 | 0.10 | 17 | 21 | 0.10 |
| ACE inhibitors | 9 | 6 | 0.12 | 8 | 6 | 0.08 |
| Angiotensin II antagonists | 43 | 34 | 0.19 | 35 | 35 | 0.003 |

ACE = angiotensin-converting enzyme; CHA$_2$DS$_2$-VASc score was based on the presence of congestive heart failure, hypertension, age ≥ 75 years, diabetes, stroke/transient ischemic attack, vascular disease, age 65–74 years, sex category (female); Charlson–Deyo index was based on the presence of myocardial infarction, congestive heart failure, peripheral vascular disease, cerebrovascular disease, dementia, chronic pulmonary disease, rheumatologic disease, peptic ulcer disease, mild liver disease, diabetes, diabetes with chronic complications, hemiplegia or paraplegia, renal disease, moderate or severe liver disease, acquired immune deficiency syndrome; COPD = chronic obstruction pulmonary disease; GI = gastrointestinal; HMG-CoA = 3-hydroxy-3-methylglutaryl coenzyme A; IPTW = inverse probability of treatment weighting; NSAID = nonsteroidal anti-inflammatory drugs; ORBIT score was based on the presence of age ≥ 74 years, anemia, bleeding history, chronic kidney disease, treatment with antiplatelet; PPI = proton pump inhibitor; SD = standard deviation; SMD = standardized mean difference.

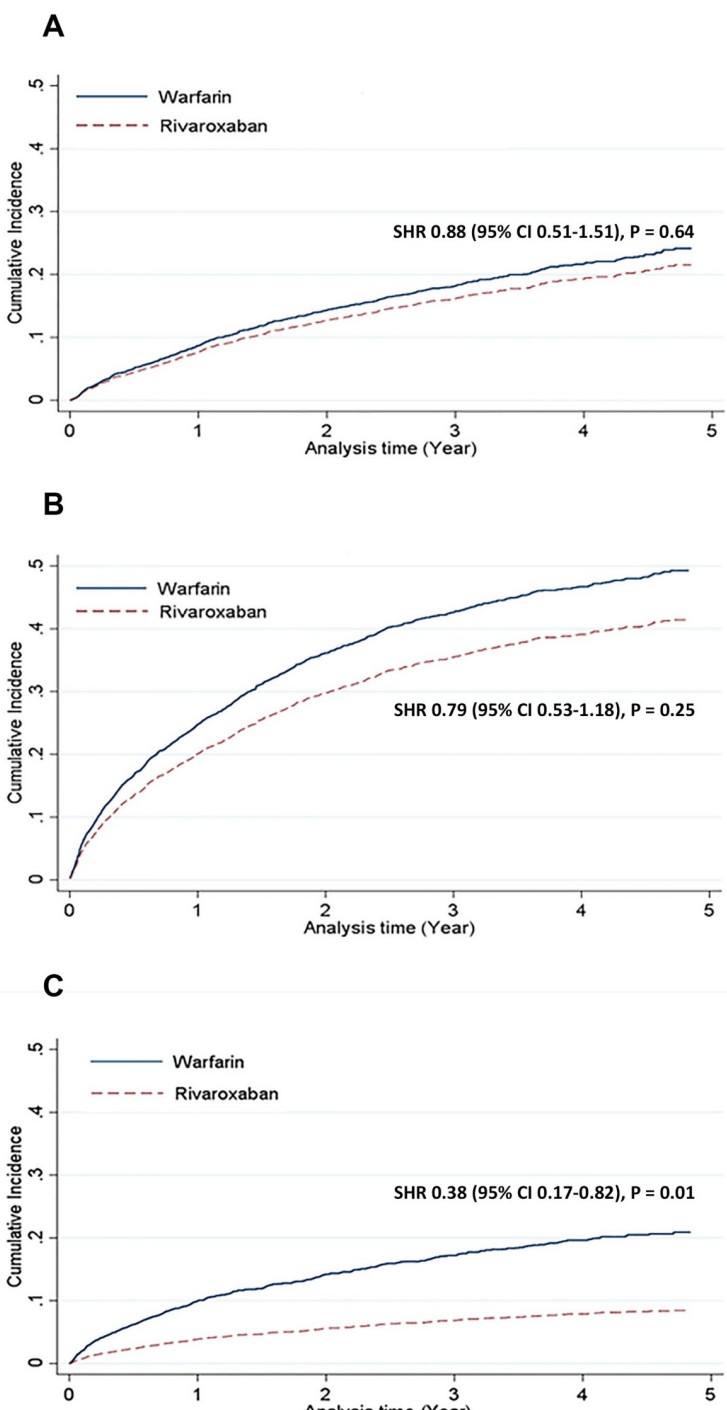

**Fig 2. Bleeding and thromboembolic outcomes: Rivaroxaban versus warfarin.** Compared with warfarin users, rivaroxaban users were associated with similar risks of major bleeding (A) and non-major clinically relevant bleeding (B), but a lower risk of ischemic stroke or systemic embolism (C). CI = confidence interval; SHR = subdistribution hazard ratio.

**Table 2. Cumulative incidence and SHR of bleeding and thromboembolic outcomes: Rivaroxaban versus warfarin.**

| Outcomes | Rivaroxaban (n = 173) | Warfarin (n = 3185) | Crude SHR (95% CI) | P-value | Adjusted SHR* (95% CI) | P-value |
|---|---|---|---|---|---|---|
| **Major bleeding** | | | | | | |
| No. of events | 23 | 560 | 0.88 (0.51–1.51) | 0.64 | 0.86 (0.50–1.47) | 0.59 |
| CICR (%) | 21.5 | 24.1 | | | | |
| **Non-major clinically relevant bleeding** | | | | | | |
| No. of events | 53 | 1267 | 0.79 (0.53–1.18) | 0.25 | 0.74 (0.48–1.13) | 0.16 |
| CICR (%) | 41.4 | 49.3 | | | | |
| **Gastrointestinal bleeding** | | | | | | |
| No. of events | 32 | 1010 | 0.62 (0.39–0.99) | 0.04 | 0.56 (0.34–0.91) | 0.02 |
| CICR (%) | 27.1 | 40.1 | | | | |
| **Intracranial bleeding** | | | | | | |
| No. of events | 7 | 236 | 0.62 (0.24–1.61) | 0.32 | 0.62 (0.24–1.61) | 0.33 |
| CICR (%) | 6.2 | 9.8 | | | | |
| **Composite endpoints of ischemic stroke or systemic embolism** | | | | | | |
| No. of events | 10 | 520 | 0.38 (0.17–0.82) | 0.01 | 0.36 (0.17–0.79) | 0.01 |
| CICR (%) | 8.4 | 20.9 | | | | |
| **Ischemic stroke** | | | | | | |
| No. of events | 7 | 236 | 0.62 (0.24–1.61) | 0.32 | 0.62 (0.24–1.61) | 0.33 |
| CICR (%) | 6.2 | 9.8 | | | | |
| **Systemic embolism** | | | | | | |
| No. of events | 6 | 311 | 0.38 (0.12–1.24) | 0.10 | 0.36 (0.11–1.12) | 0.08 |
| CICR (%) | 4.9 | 12.2 | | | | |

*Adjusted for sex, age, Charlson–Deyo index, $CHA_2DS_2$-VASc score, ORBIT score, comorbidities, and medications listed in Table 1. CI = confidence interval; CICR = cumulative incidence for competing risk; SHR = subdistribution hazard ratio.

p = 0.01), whereas the individual components of the composite endpoint were similar between the two groups.

Because 10 mg rivaroxaban 10 is approved in Taiwan for stroke prevention in NVAF patients with mild to moderate renal insufficiency, we further performed a subgroup analysis of patients receiving 10 mg rivaroxaban to evaluate its effectiveness and safety in patients with ESRD compared with warfarin (Table 3). Although no significant difference was observed in overall bleeding events (major bleeding adjusted SHR: 0.58, 95% CI: 0.24–1.37, p = 0.21; non-major clinically relevant bleeding adjusted SHR: 0.75, 95% CI: 0.40–1.39, p = 0.36), gastrointestinal bleeding risk was lower in the 10 mg rivaroxaban group (adjusted SHR: 0.43, 95% CI: 0.22–0.83, p = 0.01) than in the warfarin group. Furthermore, the composite risk of ischemic stroke or systemic embolism was significantly lower in the 10 mg rivaroxaban group than in the warfarin group (adjusted SHR: 0.32, 95% CI: 0.12–0.85, p = 0.02). The reduction in the thromboembolism risk was primarily driven by ischemic stroke (adjusted SHR: 0.31, 95% CI: 0.11–0.86, p = 0.03).

## Discussion

To our knowledge, this was the first study to evaluate the use of rivaroxaban compared with warfarin in Asian ESRD patients with NVAF using real-world data. A phase 3 randomized controlled trial comparing the clinical outcomes of rivaroxaban and warfarin included patients with moderate renal insufficiency, defined as creatinine clearance of 30–49 mL/min, and similar rates of stroke and major bleeding were observed in these patients [35]. A retrospective

**Table 3. Cumulative incidence and SHR of bleeding and thromboembolic outcomes: Rivaroxaban 10 mg versus warfarin.**

| Outcomes | Rivaroxaban 10 mg (n = 88) | Warfarin (n = 3185) | Crude SHR (95% CI) | P-value | Adjusted SHR* (95% CI) | P-value |
|---|---|---|---|---|---|---|
| **Major bleeding** | | | | | | |
| No. of events | 8 | 559 | 0.60 (0.25–1.44) | 0.25 | 0.58 (0.24–1.37) | 0.21 |
| CICR (%) | 15.2 | 24.0 | | | | |
| **Non-major clinically relevant bleeding** | | | | | | |
| No. of events | 28 | 1265 | 0.82 (0.46–1.46) | 0.49 | 0.75 (0.40–1.39) | 0.36 |
| CICR (%) | 42.4 | 49.1 | | | | |
| **Gastrointestinal bleeding** | | | | | | |
| No. of events | 13 | 1009 | 0.49 (0.25–0.95) | 0.04 | 0.43 (0.22–0.83) | 0.01 |
| CICR (%) | 22.1 | 40.1 | | | | |
| **Intracranial bleeding** | | | | | | |
| No. of events | 2 | 170 | 0.53 (0.11–2.61) | 0.44 | 0.59 (0.12–2.89) | 0.52 |
| CICR (%) | 4.2 | 7.7 | | | | |
| **Composite endpoints of ischemic stroke or systemic embolism** | | | | | | |
| No. of events | 4 | 521 | 0.33 (0.13–0.89) | 0.03 | 0.32 (0.12–0.85) | 0.02 |
| CICR (%) | 7.5 | 20.9 | | | | |
| **Ischemic stroke** | | | | | | |
| No. of events | 2 | 235 | 0.31 (0.11–0.86) | 0.02 | 0.31 (0.11–0.86) | 0.03 |
| CICR (%) | 3.1 | 9.8 | | | | |
| **Systemic embolism** | | | | | | |
| No. of events | 3 | 313 | 0.34 (0.08–1.43) | 0.14 | 0.32 (0.08–1.33) | 0.12 |
| CICR (%) | 4.3 | 12.2 | | | | |

*Adjusted for sex, age, Charlson–Deyo index, $CHA_2DS_2$-VASc score, ORBIT score, comorbidities, and medications listed in Table 1. CI = confidence interval; CICR = cumulative incidence for competing risk; SHR = subdistribution hazard ratio.

population-based US study using an ESRD database investigated the prescribing patterns and bleeding rates associated with rivaroxaban, dabigatran, and warfarin in chronic hemodialysis patients with NVAF [20]. Patients on rivaroxaban had a higher rate of major bleeding compared with those on warfarin, especially patients on 20 mg of rivaroxaban. Another retrospective cohort study in the United States compared the effectiveness and safety of rivaroxaban and warfarin in NVAF patients with stage 4 or 5 chronic kidney disease or those on hemodialysis [21]. Most patients in this study were using 20 mg of rivaroxaban. Although the risk of stroke or systemic embolism was similar between the two groups, rivaroxaban was associated with a lower rate of major bleeding compared with warfarin. These conflicting findings reflect the heterogeneity of anticoagulation responses in patients with NVAF and chronic kidney disease. In our study, we did not observe significant differences in major bleeding and non-major clinically relevant bleeding events, except lower gastrointestinal bleeding. Notably, we observed the risk of ischemic stroke or systemic embolism was lower in patients who used rivaroxaban than in those who used warfarin.

Our study revealed a high cumulative incidence of thromboembolism and bleeding in patients with ESRD on warfarin. Although warfarin has been the mainstay therapy for stroke prevention in patients with ESRD, the level of evidence is weak. According to several observational studies and meta-analyses, warfarin is associated with an increased bleeding risk, including intracranial hemorrhage, in patients with ESRD, without providing a protective effect against stroke [6, 9–12, 36]. In addition, the risk of warfarin-related major bleeding, particularly intracranial bleeding, is significantly higher in Asian populations [37]. Our study did not include patients with no anticoagulation because such a study design imposes a significant

confounding effect that cannot be eliminated through statistical adjustments. Furthermore, direct oral anticoagulants have fewer drug and food interactions and monitoring requirements. In addition, some studies have indicated that factor Xa inhibition may ameliorate nephropathy, and rivaroxaban has been associated with a slower decline in renal function compared with warfarin [38, 39]. A pharmacokinetic study of rivaroxaban in chronic hemodialysis patients indicated that 10 mg of the drug administered to such patients had similar outcomes to 20 mg given to healthy volunteers [40], and drug accumulation was absent after multiple doses. Our study showed that 10 mg of rivaroxaban was associated with a lower rate of thromboembolism and gastrointestinal bleeding compared with warfarin, but the use of this dose in patients with NVAF and ESRD requires further investigation, considering the retrospective nature of our study. Other therapies such as antiplatelet drugs have been investigated in retrospective observational studies, and they did not confer stroke prevention benefits in patients with ESRD [41, 42]. The current guidelines do not recommend antiplatelet agent use for stroke prevention [43]. Therefore, our study did not compare clinical outcomes between rivaroxaban and antiplatelet agents.

The present study has several limitations. First, the data source was insurance claims, and thus, information on international normalized ratio levels was unavailable; for this reason, we could not evaluate the time in the therapeutic range of the warfarin group. Information of actual adherence rates were unavailable in our data source. Second, the sample size of the rivaroxaban group was small because rivaroxaban was used off-label. Further studies with a larger Asian ESRD population are required to evaluate the effectiveness and safety of rivaroxaban. Finally, the retrospective study design and differences between rivaroxaban and warfarin users at baseline indicated the potential for confounding effects. Therefore, we used IPTW to minimize the effect of confounding variables, and our analyses were adjusted for baseline characteristics.

## Conclusions

In real-world clinical settings, warfarin is associated with a high incidence of thromboembolic and bleeding events in Taiwanese patients with NVAF and ESRD. Rivaroxaban use in this population resulted in fewer thrombotic events but similar major bleeding events when compared with warfarin. A prospective clinical study is required to confirm the findings of the present study.

## Supporting information

**S1 Table. Disease diagnosis codes according to ICD-9-CM, ATC classification of medications, and reimbursement codes for procedures.**
(DOCX)

## Acknowledgments

We appreciate the assistance provided by Health and Clinical Data Research Center, College of Public Health, Taipei Medical University, Taipei, Taiwan.

## Author Contributions

**Conceptualization:** Yi-Cheng Lin, Chun-Ming Shih, Chih-Wei Chen, Chien-Yi Hsu, Te-Chao Fang, Chun-Yao Huang.

**Data curation:** Yi-Cheng Lin, Chih-Wei Chen, Chien-Yi Hsu, Li-Ying Chen, Li-Nien Chien.

**Formal analysis:** Yi-Cheng Lin, Li-Ying Chen, Li-Nien Chien, Te-Chao Fang.

**Funding acquisition:** Bi-Li Chen.

**Investigation:** Yi-Cheng Lin, Chih-Wei Chen, Yung-Ta Kao, Li-Nien Chien, Te-Chao Fang, Chun-Yao Huang.

**Methodology:** Yi-Cheng Lin, Li-Nien Chien, Te-Chao Fang, Chun-Yao Huang.

**Project administration:** Bi-Li Chen, Chun-Ming Shih, Feng-Yen Lin, Chun-Yao Huang.

**Resources:** Bi-Li Chen, Chun-Ming Shih, Feng-Yen Lin, Chien-Yi Hsu, Chun-Yao Huang.

**Supervision:** Bi-Li Chen, Chun-Ming Shih, Feng-Yen Lin, Yung-Ta Kao, Wei-Fung Bi, Li-Nien Chien, Te-Chao Fang, Chun-Yao Huang.

**Validation:** Yi-Cheng Lin, Bi-Li Chen, Chun-Ming Shih, Feng-Yen Lin, Chien-Yi Hsu, Yung-Ta Kao, Wei-Fung Bi, Li-Nien Chien, Te-Chao Fang, Chun-Yao Huang.

**Visualization:** Yi-Cheng Lin, Wei-Fung Bi, Li-Ying Chen, Chun-Yao Huang.

**Writing – original draft:** Yi-Cheng Lin, Li-Nien Chien, Chun-Yao Huang.

**Writing – review & editing:** Yi-Cheng Lin, Chun-Ming Shih, Li-Nien Chien, Chun-Yao Huang.

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
