## [Decision Letter · Decision Letter 0]

22 Jan 2021

PONE-D-20-36593

Effectiveness and safety of rivaroxaban versus warfarin in Taiwanese patients with end-stage renal disease and non-valvular atrial fibrillation: A real-world nationwide cohort study

PLOS ONE

Dear Dr. Huang,

Thank you for submitting your manuscript to PLOS ONE. After careful consideration, we feel that it has merit but does not fully meet PLOS ONE’s publication criteria as it currently stands. Therefore, we invite you to submit a revised version of the manuscript that addresses the points raised during the review process.

We look forward to receiving your revised manuscript.

Kind regards,

Yoshihiro Fukumoto

Academic Editor

PLOS ONE

https://www.sciencedirect.com/science/article/pii/S0735109718349982?via%3Dihub

In your revision ensure you cite all your sources (including your own works), and quote or rephrase any duplicated text outside the methods section. Further consideration is dependent on these concerns being addressed.

Reviewers' comments:

Reviewer's Responses to Questions

**Comments to the Author**

1. Is the manuscript technically sound, and do the data support the conclusions?

Reviewer #1: Yes

Reviewer #2: Yes

2. Has the statistical analysis been performed appropriately and rigorously? 

Reviewer #1: Yes

Reviewer #2: Yes

3. Have the authors made all data underlying the findings in their manuscript fully available?

Reviewer #1: No

Reviewer #2: Yes

4. Is the manuscript presented in an intelligible fashion and written in standard English?

Reviewer #1: Yes

Reviewer #2: Yes

5. Review Comments to the Author

Reviewer #1: Huang et al. conducted a retrospective observational cohort study using a database of the national health insurance in Taiwan, elucidating the effectiveness and safety of rivaroxaban, in comparison to warfarin, in patients with ESRD and NVAF. There were significantly less thromboembolic episodes in patients taking rivaroxaban whereas bleeding events were comparable between the groups. The authors concluded that rivaroxaban may be associated with a lower risk of thromboembolism compared to warfarin in patients with ESRD and NVAF. The number of studied patient was relatively small and data with regard to drug adherence is not available. There are apparent limitations in the present study due to its study design and data source, although clinically important issues are dealt with.

Major comments:

#1. Setting of the control group:

As the author stated in the manuscript, warfarin therapy is not established in the studied population. Data of patients with ESRD and NVAF NOT taking anticoagulation is important and should be compared. In addition, TTR in the warfarin group was not presented. Is this data unavailable?

#2. Eligible patients:

In the method section, the authors stated that eligible patient were selected by the use of erythropoiesis-stimulating agents, which was limited to ESRD patients. The situation is understandable, however, how can it be proven that they truly extracted patients with ESRD? How do they exclude the possibility of over-indication of erythropoiesis-stimulating agents?

#3. What was the proportion of patients with hemodialysis?

#4. Issue of drug adherence:

The authors should state about this in the discussion and limitations.

Minor comments:

#1. Has the number of patient unchanged after the IPTW (table 1)?

#2. Number of patient receiving 10mg rivaroxaban should be stated in the text and table 3.

#3. The manuscript is basically well written. However, English usage should be revised through the manuscript again.

Reviewer #2: Comments to the Author

Yi-Cheng Lin et al. examined the efficacy and safety of rivaroxaban and warfarin in patients with end-stage renal disease in a population-based cohort study.

The author has clearly shown valuable information that will change future clinical practice. I think this paper is a well-written and informative study.

Major comments

This paper is an important study analyzed in detail by population-based cohort study using National Health Insurance.

If possible, please consider the content of the following papers.

Biol Pharm Bull 2011;34:824-30: The role of PAR2 in the progression of renal function.(This paper is related to the other fact that type Xa DOAC prevents the role of PAR2)

In addition, as shown in Reference paper 20, comparing with antiplatelet drugs and examining dialysis cases leads to deeper consideration..

6. PLOS authors have the option to publish the peer review history of their article (what does this mean?). If published, this will include your full peer review and any attached files.

Reviewer #1: No

Reviewer #2: No

---

## [Author Response · Author response to Decision Letter 0]

26 Feb 2021

Reviewer #1: 

1. Setting of the control group: As the author stated in the manuscript, warfarin therapy is not established in the studied population. Data of patients with ESRD and NVAF NOT taking anticoagulation is important and should be compared.

Authors’ response: Thank you for your valuable suggestions. While comparing rivaroxaban with no anticoagulation in patients with ESRD and NVAF is important, there is a concern of significant confounding effect (confounding by indication) in such study design. For example, clinicians may choose not to use anticoagulants in patients with high bleeding risk, and such risk may not be captured by claims data. Other publication regarding comparison of direct oral anticoagulant and warfarin also raised similar issue (Circulation. 2018;138:1519–1529). Prospective study is needed to answer this important question. According to your comments, we revised the Discussion section of the manuscript as below (page 23 line 225):

Our study did not include patients with no anticoagulation because such a study design imposes a significant confounding effect that cannot be eliminated through statistical adjustments.

2. TTR in the warfarin group was not presented. Is this data unavailable?

Authors’ response: TTR in the warfarin group is not available since our data source does not contain laboratory data. This was included as one of our study limitations (page 24 line 242).

3. Eligible patients: In the method section, the authors stated that eligible patient were selected by the use of erythropoiesis-stimulating agents, which was limited to ESRD patients. The situation is understandable, however, how can it be proven that they truly extracted patients with ESRD? How do they exclude the possibility of over-indication of erythropoiesis-stimulating agents?

Authors’ response: Thank you for your comments. In order to eliminate possibility of over-indication of erythropoiesis-stimulating agents (ESA), we only included ESA users who also had end-stage renal disease (ESRD) diagnosis. To avoid misunderstanding, we revised the Methods section of the manuscript as below (page 8 line 67):

For erythropoiesis-stimulating drug users, we specifically included those with an ESRD diagnosis identified through ICD-9 codes to eliminate patients using these agents for off-label indications.

4. What was the proportion of patients with hemodialysis?

Authors’ response: The proportion of patients on dialysis was 82.5%, with 67 subjects in the rivaroxaban group and 2702 subjects in the warfarin group. We did not perform analyses in patients on dialysis separately because rivaroxaban cannot be eliminated by renal replacement therapy. Drug concentration is likely similar between ESRD patients who are not on dialysis and patients who are on regular dialysis.

5. Issue of drug adherence: The authors should state about this in the discussion and limitations.

Authors’ response: Thank you for your comments. Since the actual adherence rate cannot be determined due to the nature of our data source, we revised the Discussion section of the manuscript as below (Page 24 line 243):

Information of actual adherence rates were unavailable in our data source. 

6. Has the number of patient unchanged after the IPTW (table 1)?

Authors’ response: The purpose of IPTW is to create a synthetic sample in which treatment assignment is independent of baseline covariates (Stat Med. 2015 Dec 10; 34(28): 3661–3679). Thus, the number of patients might slightly change after IPTW. The advantage of this method is that it uses the entire cohort rather than selecting some matching subjects. We used this method in our previous published work (J Am Coll Cardiol. 2018 Jul 31;72(5):477-485).

7. Number of patient receiving 10mg rivaroxaban should be stated in the text and table 3.

Authors’ response: Number of patients receiving 10 mg rivaroxaban was stated in the Result section of the manuscript (page 12 line 126). The number was added to Table 3 according to your comment.

8. The manuscript is basically well written. However, English usage should be revised through the manuscript again.

Authors’ response: Thank you for your advice. We submitted our manuscript to English editing service. Revised manuscript with track changes is attached.

Reviewer #2: 

1. If possible, please consider the content of the following papers: Biol Pharm Bull 2011;34:824-30: The role of PAR2 in the progression of renal function.(This paper is related to the other fact that type Xa DOAC prevents the role of PAR2)

Authors’ response: Thank you for providing this important paper. We added this paper into the Discussion section of the manuscript as below (page 23 line 227):

Furthermore, direct oral anticoagulants have fewer drug and food interactions and monitoring requirements. In addition, some studies have indicated that factor Xa inhibition may ameliorate nephropathy, and rivaroxaban has been associated with a slower decline in renal function compared with warfarin [38, 39].

2. In addition, as shown in Reference paper 20, comparing with antiplatelet drugs and examining dialysis cases leads to deeper consideration.

Authors’ response: The use of antiplatelet drugs in end-stage renal disease (ESRD) patients with atrial fibrillation for stroke prevention has been seen in clinical practice. However, clinical outcomes of antiplatelet drugs in this population has been unfavorable and is not recommended by clinical guidelines. Therefore, we did not compare rivaroxaban with antiplatelet drugs in the present study. In light of your valuable comments, we revised the Discussion section of the manuscript as below (page 24 line 236):

Other therapies such as antiplatelet drugs have been investigated in retrospective observational studies, and they did not confer stroke prevention benefits in patients with ESRD [41, 42]. The current guidelines do not recommend antiplatelet agent use for stroke prevention [43]. Therefore, our study did not compare clinical outcomes between rivaroxaban and antiplatelet agents.

---

## [Decision Letter · Decision Letter 1]

29 Mar 2021

Effectiveness and safety of rivaroxaban versus warfarin in Taiwanese patients with end-stage renal disease and nonvalvular atrial fibrillation: A real-world nationwide cohort study

PONE-D-20-36593R1

Dear Dr. Huang,

We’re pleased to inform you that your manuscript has been judged scientifically suitable for publication and will be formally accepted for publication once it meets all outstanding technical requirements.

Kind regards,

Yoshihiro Fukumoto

Academic Editor

PLOS ONE

Additional Editor Comments (optional):

Reviewers' comments:

Reviewer's Responses to Questions

**Comments to the Author**

1. If the authors have adequately addressed your comments raised in a previous round of review and you feel that this manuscript is now acceptable for publication, you may indicate that here to bypass the “Comments to the Author” section, enter your conflict of interest statement in the “Confidential to Editor” section, and submit your "Accept" recommendation.

Reviewer #1: All comments have been addressed

Reviewer #2: All comments have been addressed

2. Is the manuscript technically sound, and do the data support the conclusions?

Reviewer #1: Yes

Reviewer #2: Yes

3. Has the statistical analysis been performed appropriately and rigorously? 

Reviewer #1: Yes

Reviewer #2: Yes

4. Have the authors made all data underlying the findings in their manuscript fully available?

Reviewer #1: Yes

Reviewer #2: Yes

5. Is the manuscript presented in an intelligible fashion and written in standard English?

Reviewer #1: Yes

Reviewer #2: Yes

6. Review Comments to the Author

Reviewer #1: The manuscript has been much improved. The study results will be informative for readers of the journal. The reviewer has no further comments.

Reviewer #2: Comments to the Author

This manuscript has been improved very much.

This paper is acceptable for publication as is.

7. PLOS authors have the option to publish the peer review history of their article (what does this mean?). If published, this will include your full peer review and any attached files.

Reviewer #1: **Yes: **Yasushi Mukai, MD, PhD

Reviewer #2: No

---

## [Editor Report · Acceptance letter]

31 Mar 2021

PONE-D-20-36593R1 

Effectiveness and safety of rivaroxaban versus warfarin in Taiwanese patients with end-stage renal disease and nonvalvular atrial fibrillation: A real-world nationwide cohort study 

Dear Dr. Huang:

I'm pleased to inform you that your manuscript has been deemed suitable for publication in PLOS ONE. Congratulations! Your manuscript is now with our production department. 

Kind regards, 

on behalf of

Dr. Yoshihiro Fukumoto 

Academic Editor

PLOS ONE